# PettingZoo: A Standard API for Multi-Agent Reinforcement Learning

**J. K. Terry**[*][†]
j.k.terry@swarmlabs.com

**Benjamin Black**[*][†]
benjamin.black@swarmlabs.com

**Nathaniel Grammel**[†]
ngrammel@umd.edu

**Mario Jayakumar**[†]
mariojay@umd.edu

**Ananth Hari**[‡]
ahari1@umd.edu

**Ryan Sullivan**[*][†]
ryan.sullivan@swarmlabs.com

**Luis Santos**[§]
lss@umd.edu

**Rodrigo Perez**[¶]
rlazcano@umd.edu

**Caroline Horsch**[*][†]
caroline.horsch@swarmlabs.com

**Clemens Dieffendahl**[‖]
dieffendahl@campus.tu-berlin.de

**Niall L. Williams**[†]
niallw@umd.edu

**Yashas Lokesh**[†]
yashloke@umd.edu

**Praveen Ravi**[†]
pravi@umd.edu

## Abstract

This paper introduces the PettingZoo library and the accompanying Agent Environment Cycle ("AEC") games model. PettingZoo is a library of diverse sets of multi-agent environments with a universal, elegant Python API. PettingZoo was developed with the goal of accelerating research in Multi-Agent Reinforcement Learning ("MARL"), by making work more interchangeable, accessible and reproducible akin to what OpenAI's Gym library did for single-agent reinforcement learning. PettingZoo's API, while inheriting many features of Gym, is unique amongst MARL APIs in that it's based around the novel AEC games model. We argue, in part through case studies on major problems in popular MARL environments, that the popular game models are poor conceptual models of games commonly used in MARL and accordingly can promote confusing bugs that are hard to detect, and that the AEC games model addresses these problems.

## 1 Introduction

Multi-Agent Reinforcement Learning (MARL) has been behind many of the most publicized achievements of modern machine learning — AlphaGo Zero [Silver et al., 2017], OpenAI Five [OpenAI, 2018], AlphaStar [Vinyals et al., 2019]. These achievements motivated a boom in MARL research, with Google Scholar indexing 9,480 new papers discussing multi-agent reinforcement learning in 2020 alone. Despite this boom, conducting research in MARL remains a significant engineering

---

[*]Swarm Labs

[†]Department of Computer Science | University of Maryland, College Park

[‡]Department of Electrical and Computer Engineering | University of Maryland, College Park

[§]Department of Mechanical Engineering | University of Maryland, College Park

[¶]Maryland Robotics Center | University of Maryland, College Park

[‖]Faculty of Electrical Engineering and Computer Science | Technical University of Berlin

35th Conference on Neural Information Processing Systems (NeurIPS 2021).

challenge. A large part of this is because, unlike single agent reinforcement learning which has OpenAI's Gym, no de facto standard API exists in MARL for how agents interface with environments. This makes the reuse of existing learning code for new purposes require substantial effort, consuming researchers' time and preventing more thorough comparisons in research. This lack of a standardized API has also prevented the proliferation of learning libraries in MARL. While a massive number of Gym-based single-agent reinforcement learning libraries or code bases exist (as a rough measure 669 pip-installable packages depend on it at the time of writing GitHub [2021]), only 5 MARL libraries with large user bases exist [Lanctot et al., 2019, Weng et al., 2020, Liang et al., 2018, Samvelyan et al., 2019, Nota, 2020]. The proliferation of these Gym based learning libraries has proved essential to the adoption of applied RL in fields like robotics or finance and without them the growth of applied MARL is a significantly greater challenge. Motivated by this, this paper introduces the PettingZoo library and API, which was created with the goal of making research in MARL more accessible and serving as a multi-agent version of Gym.

Prior to PettingZoo, the numerous single-use MARL APIs almost exclusively inherited their design from the two most prominent mathematical models of games in the MARL literature—Partially Observable Stochastic Games ("POSGs") and Extensive Form Games ("EFGs"). During our development, we discovered that these common models of games are not conceptually clear for multi-agent games implemented in code and cannot form the basis of APIs that cleanly handle all types of multi-agent environments.

To solve this, we introduce a new formal model of games, Agent Environment Cycle ("AEC") games that serves as the basis of the PettingZoo API. We argue that this model is a better conceptual fit for games implemented in code. and is uniquely suitable for general MARL APIs. We then prove that any AEC game can be represented by the standard POSG model, and that any POSG can be represented by an AEC game. To illustrate the importance of the AEC games model, this paper further covers two case studies of meaningful bugs in popular MARL implementations. In both cases, these bugs went unnoticed for a long time. Both stemmed from using confusing models of games, and would have been made impossible by using an AEC games based API.

The PettingZoo library can be installed via `pip install pettingzoo`, the documentation is available at `https://www.pettingzoo.ml`, and the repository is available at `https://github.com/Farama-Foundation/PettingZoo`.

## 2 Background and Related Works

Here we briefly survey the state of modeling and APIs in MARL, beginning by briefly looking at Gym's API (Figure 1). This API is the de facto standard in single agent reinforcement learning, has largely served as the basis for subsequent multi-agent APIs, and will be compared to later.

```python
import gym
env = gym.make('CartPole-v0')
observation = env.reset()
for _ in range(1000):
    action = policy(observation)
    observation, reward, done, info = env.step(action)
```

Figure 1: An example of the basic usage of Gym

```python
from ray.rllib.examples.env.multi_agent
    import MultiAgentCartPole
env = MultiAgentCartPole()
observation = env.reset()
for _ in range(1000):
    actions = policies(agents, observation)
    observation, rewards, dones,
        infos = env.step(actions)
```

Figure 2: An example of the basic usage of RLlib

The Gym API is a fairly straightforward Python API that borrows from the POMDP conceptualization of RL. The API's simplicity and conceptual clarity has made it highly influential, and it naturally accompanying the pervasive POMDP model that's used as the pervasive mental and mathematical model of reinforcement learning [Brockman et al., 2016]. This makes it easier for anyone with an understanding of the RL framework to understand Gym's API in full.

### 2.1 Partially Observable Stochastic Games and RLlib

Multi-agent reinforcement learning does not have a universal mental and mathematical model like the POMDP model in single-agent reinforcement learning. One of the most popular models is the partially observable stochastic game ("POSG"). This model is very similar to, and strictly more

general than, multi-agent MDPs [Boutilier, 1996], Dec-POMDPs [Bernstein et al., 2002], and Stochastic ("Markov") games [Shapley, 1953]). In a POSG, all agents step together, observe together, and are rewarded together. The full formal definition is presented in Appendix C.1

This model of simultaneous stepping naturally translates into Gym-like APIs, where the actions, observations, rewards, and so on are lists or dictionaries of individual values for agents. This design choice has become the standard for MARL outside of strictly turn-based games like poker, where simultaneous stepping would be a poor conceptual fit [Lowe et al., 2017, Zheng et al., 2017, Gupta et al., 2017, Liu et al., 2019, Liang et al., 2018, Weng et al., 2020]. One example of this is shown in Figure 2 with the multi-agent API in RLlib [Liang et al., 2018], where agent-keyed dictionaries of actions, observations and rewards are passed in a simple extension of the Gym API.

This model has made it much easier to apply single agent RL methods to multi-agent settings. However, there are two immediate problems with this model:

1. Supporting strictly turn-based games like chess requires constantly passing dummy actions for non-acting agents (or using similar tricks).
2. Changing the number of agents for agent death or creation is very awkward, as learning code has to cope with lists suddenly changing sizes.

## 2.2 OpenSpiel and Extensive Form Games

In the cases of strictly turn based games where POSG models are poorly suited (e.g. Chess), MARL researchers generally mathematically model the games as Extensive Form Games ("EFG"). The EFG represents games as a tree, *explicitly* representing every possible sequence of actions as a root to leaf path in the tree. Stochastic aspects of a game (or MARL environment) are captured by adding a "Nature" player (sometimes also called "Chance") which takes actions according to some given probability distribution. For a full definition of EFGs, we refer the reader to Osborne and Rubinstein [1994] or Appendix C.2. OpenSpiel [Lanctot et al., 2019], a major library with a large collection of classical board and card games for MARL bases their API off of the EFG paradigm, the API of which is shown in Figure 3.

```python
import pyspiel
import numpy as np

game = pyspiel.load_game("kuhn_poker")
state = game.new_initial_state()
while not state.is_terminal():
    if state.is_chance_node():
        # Step the stochastic environment.
        action_list, prob_list = zip(*state.chance_outcomes())
        state.apply_action(np.random.choice(action_list, p=prob_list))
    else:
        # sample an action for the agent
        legal_actions = state.legal_actions()
        observations = state.observation_tensor()
        action = policies(state.current_agent(), legal_actions, observations)
        state.apply_action(action)
        rewards = state.rewards()
```

Figure 3: An example of the basic usage of OpenSpiel

The EFG model has been successfully used for solving problems involving theory of mind with methods like game theoretic analysis and tree search. However, for application in general MARL problems, three immediate concerns arise with the EFG model:

1. The model, and the corresponding API, is very complex compared to that of POSGs, and isn't suitable for beginners the way Gym is—this environment API is much more complicated than Gym's API or RLLib's POSG API for example. Furthermore, due to the complexity of the EFG model, reinforcement learning researchers don't ubiquitously use it as a mental model of games in the same way that they use the POSG or POMDP model.
2. The formal definition only includes rewards at the end of games, while reinforcement learning often requires frequent rewards. While this is possible to work around in the API implementation, it is not ideal.

3. The OpenSpiel API does not handle continuous actions (a common and important case in RL), though this was a choice that is not inherent to the EFG model.

It's also worth briefly noting that some simple strictly turn based games are modeled with the single-agent Gym API, with the environment alternating which agent is controlled, [Ha, 2020]. This approach is unable to reasonably scale beyond two agents due to the difficulties of handling changes in agent order (e.g. Uno), agent death, and agent creation.

# 3 PettingZoo Design Goals

Our development of PettingZoo both as a general library and an API centered around the following goals.

## 3.1 Be like Gym

In PettingZoo, we wanted to leverage Gym's ubiquity, simplicity and universality. This created two concrete goals for us:

- Make the API look and feel like Gym, and relatedly make the API pythonic and simple
- Include numerous reference implementations of games with the main package

Reusing as many design metaphors from Gym as possible will help its massive existing user base to almost instantly understand PettingZoo's API. Similarly, for an API to become standardized, it must support a large collection of useful environments to attract users and for adoption to begin, similar to what Gym did.

## 3.2 Be a Universal API

If there is to be a Gym-like API for MARL, it has to be able to support all use cases and types of environments. Accordingly, several technically difficult cases exist that have to be carefully considered:

- Environments with large numbers of agents
- Environments with agent death and creation
- Environments where different agents can be chosen to participate in each episode
- Learning methods that require access to specialty low level features

Two related softer design goals for universal design are ensuring the API is simple enough for beginners to easily use, and making the API easily changeable if the direction of research in the field dramatically changes.

# 4 Case Studies of Problems With The POSG Model in MARL

To supplement the description of the problems with the POSG models described in Section 2.1, we overview problems with basing APIs around these models that could theoretically occur in software games, and then examine real cases of those problems occurring in popular MARL environments. We specifically focus on POSGs here because EFG based APIs are extraordinarily rare (OpenSpiel is the only major one), while POSG based ones are almost universal.

## 4.1 POSGs Don't Allow Access To Information You Should Have

Another problem with modeling environments using simultaneous actions in the POSG model is that all of an agent's rewards (from all sources) are summed and returned all at once. In a multi-agent game though, this combined reward is often the composite reward from the actions of other agentss and the environment. Similarly, you might want to be able to attribute the source of this reward for various learning reasons, or for debugging purposes to find out the origin of your rewards. However, in thinking about reward origins, having all rewards emitted at once proves to be very confusing

because rewards from different sources are all combined. Accessing this information via an API modeled after a POSG requires deviating from the model. This would come in the form of returning a 2D array of rewards instead of a list, which would be difficult to standardize and inconvenient for learning code to parse.

A notable case where this caused an issue in practice is in the popular pursuit gridworld environment from Gupta et al. [2017], shown in Figure 4. In it, 8 red controllable pursuer must work together to surround and capture 30 randomly moving blue evaders. The action space of each pursuer is discrete (cardinal directions or do nothing), and the observation space is a $7 \times 7$ box centered around a pursuer (depicted by the orange box). When an evader is surrounded on all sides by pursuers or the game boundaries, each contributing pursuer gets a reward of 5.

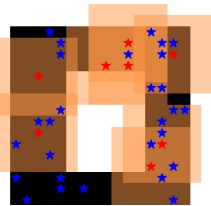

Figure 4: The *pursuit* environment from Gupta et al. [2017].

In pursuit, pursuers move first, and then evaders move randomly, before it's determined if an evader is captured and rewards are emitted. Thus an evader that "should have" been captured is not actually captured. Having the evaders move second isn't a bug, it's just way of adding complexity to the classic genre of pursuer/evader multi-agent environments [Vidal et al., 2002], and is representative of real problems. When *pursuit* is viewed as an AEC game, we're forced to attribute rewards to individual steps, and the breakdown becomes pursuers receiving deterministic rewards from surrounding the evader, and then random reward due to the evader moving after. Removing this random component of the reward (the part caused by the evaders action after the pursuers had already moved), should then lead to superior performance. In this case the problem was so innocuous that fixing it required switching two lines of code where their order made no obvious difference. We experimentally validate this performance improvement in Appendix A.1, showing that on average this change resulted in up to a 22% performance in the expected reward of a learned policy.

Bugs of this family could easily happen in almost any MARL environment, and analyzing and preventing them is made much easier when using the POSG model. Because every agent's rewards are summed together in the POSG model, this specific problem when looking at the code was extraordinarily non-obvious, whereas when forced to attribute the reward of individual agents this becomes clear. Moreover if an existing environment had this problem, by exposing the actual sources of rewards to learning code researchers are able to remove differing sources of reward to more easily find and remove bugs like this, and in principle learning algorithms could be developed that automatically differently weighted different sources of reward.

### 4.2 POSGs Based APIs Are Not Conceptually Clear For Games Implemented In Code

Introducing race conditions is a very easy mistake to make in MARL code in practice, and this occurs because simultaneous models of multi-agent games are not representative of how game code normally executes. This stems from a very common scenario in multi-agent environments where two agents are able to take conflicting actions (i.e. moving into the same space). This discrepancy has to be resolved by the environment (i.e. collision handling); which we call "tie-breaking."

Consider an environment with two agents, Alice and Bob, in which Alice steps first and tie-breaking is biased in Alice's favor. If such an environment were assumed to have simultaneous actions, then observations for both agents would be taken before either acted, causing the observation Bob acts on to no longer be an accurate representation of the environment if a conflict with biased tie-breaking occurs. For example, if both agents tried to step into the same square and Alice got the square because she was first in the list, Bob's observation before acting was effectively inaccurate and the environment was not truly parallel. This behavior is a true race condition—the result of stepping through the environment can inadvertently differ depending on the internal resolution order of agent actions.

In any environment that's even slightly complex, a tremendous number of instances where tie-breaking must be handled will typically occur. In any cases where a single one is missed, the environment will have race conditions that your code will attempt to learn. While finding these will always be important, a valuable tool to mitigate these possibilities is to use an API that treats each agent as acting sequentially, returning new observations afterwards. This entirely prevents the opportunity for introducing race conditions. Moreover, this entire problem stems from the fact that using APIs that model agents as updating sequentially for software MARL environments generally makes more conceptual sense than modeling the updates as simultaneous—unless the authors of environments use very complex parallelization, the environments will *actually* be updated one agent at a time. It is worth mentioning that this race condition cannot occur in an environment simulated in the physical world with continuous time or a simulated environment with a sufficient amount of observation delay (though most actively researched environment in MARL do not currently have any observation delay).

In Appendix A.1 we go through a case study of a race condition like this happening in the open source implementation of the social sequential dilemma game environments [Vinitsky et al., 2019]. These are popular multi-agent grid world environments intended to study emergent behaviors for various forms of resource management, and has imperfect tie-breaking in a case where two agents try to act on resources in the same grid while using a simultaneous API. This bug in particular illustrates how extraordinarily difficult making all tie-breaking truly unbiased is in practice even for fairly simple environments. We defer this to the appendix as explaining the specific origin requires a large amount of exposition and diagrams about the rules of the environment.

## 5 The Agent Environment Cycle Games Model

Motivated by the problems with applying the POSG and EFG models to MARL APIs, we developed the Agent Environment Cycle ("AEC") Game. In this model, agents sequentially see their observation, agents take actions, rewards are emitted from the other agents, and the next agent to act is chosen. This is effectively a sequentially stepping form of the POSG model.

Modeling multi-agent environments sequentially for APIs has numerous benefits:

- It allows for clearer attribution of rewards to different origins, allowing for various learning improvements, as described in Section 4.1.
- It prevents developers adding confusing and easy-to-introduce race conditions, as described in Section 4.2.
- It more closely models how computer games are executed in code, as described in Section 4.2.
- It formally allows for rewards after every step as is required in RL, but is not generally a part of the EFG model, as discussed in Section 2.2.
- It is simple enough to serve as a mental model, especially for beginners, unlike the EFG model as discussed in Section 2.2 and illustrated in the definition in Appendix C.2.
- Changing the number of agents for agent death or creation is less awkward, as learning code does not have to account for lists constantly changing sizes, as discussed in Section 2.1.
- It is the least bad option for a universal API, compared to simultaneous stepping, as alluded to in Section 2.1. Simultaneous stepping requires the use of no-op actions if not all agents can act which are very difficult to deal with, whereas sequentially stepping agents that could all act simultaneously and queuing up their actions is not especially inconvenient.

In Appendix C.3 we mathematically formalize the AEC games model, however understanding the formalism in full is not essential to understanding the paper. In Appendix D we further prove that for every AEC game an equivalent POSG exists and that for every POSG an equivalent AEC game exists. This shows that the AEC games model is as powerful a model as the most common current model of multi-agent environments.

One additional conceptual feature of the AEC games model exists that we have not previously discussed because it does not usually play a role in APIs (see Section 6.4). In the AEC games model, we deviate from the POSG model by introducing the "environment" agent, which is analogous to the Nature agent from EFGs. When this agent acts in the model it indicates the updating of

the environment itself, realizing and reacting to submitting agent actions. This allows for a more comprehensive attribution of rewards, causes of agent death, and discussion of games with strange updating rules and race conditions. An example of the transitions for Chess is shown in Figure 5, which serves as the inspiration for the name "agent environment cycle".

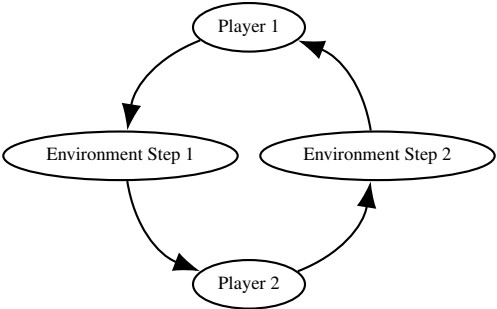

Figure 5: The AEC diagram of Chess

# 6  API Design

## 6.1  Basic API

The PettingZoo API is shown in Figure 6, and the strong similarities to the Gym API (Figure 1) should be obvious — each agent provides an `action` to a `step` function and receives `observation`, `reward`, `done`, `info` as the return values. The observation and state spaces also use the the exact same space objects as Gym. The `render` and `close` methods also function identically to Gym's, showing a current visual frame representing the environment to the screen whenever called. The `reset` method similarly has identical function to Gym — it resets the environment to a starting configuration after being played through. PettingZoo really only has two deviations from the regular Gym API — the `last` and `agent_iter` methods and the corresponding iteration logic.

```
from pettingzoo.butterfly import pistonball_v0
env = pistonball_v0.env()
env.reset()
for agent in env.agent_iter(1000):
    env.render()
    observation, reward, done, info = env.last()
    action = policy(observation, agent)
    env.step(action)
env.close()
```

Figure 6: An example of the basic usage of Pettingzoo

## 6.2  The `agent_iter` Method

The `agent_iter` method is a generator method of an environment that returns the next agent that the environment will be acting upon. Because the environment is providing the next agent to act, this cleanly abstracts away any issues surrounding changing agent orders, agent generation, and agent death. This generation also parallels the functionality of the next agent function from the AEC games model. This method, combined with one agent acting at once, allows for the support of every conceivable variation of the set of agents changing.

## 6.3  The `last` Method

An odd aspect of multi-agent environments is that from the perspective of one agent, the other agents are part of the environment. Whereas in the single agent case the observation and rewards can be given immediately, in the multi-agent case an agent has to wait for all other agents to act before it's `observation`, `reward`, `done` and `info` can be fully determined. For this reason, these values are given by the `last` method, and they can then be passed into a policy to choose an action. Less robust

implementations would not allow for features like changing agent orders (like the reverse card in Uno).

## 6.4  Additional API Features

The `agents` attribute is a list of all agents in the environment, as strings. The `rewards`, `dones`, `infos` attributes are agent-keyed dictionaries for each attribute (note that the rewards are the instantaneous ones resulting from the most recent action). These allow access to agent properties at all points on a trajectory, regardless of which is selected. The `action_space(agent)` and `observation_space(agent)` functions return the static action and observation spaces respectively for the agent given as an argument. The `observe(agent)` method provides the observation for a single agent by passing its name as an argument, which can be useful if you need to observe an agent in an unusual context. The `state` method is an optional method returns the global state of an environment, as is required for centralized critic methods. The `agent_selection` method returns the agent that can currently be acted upon per `agent_iter`.

The motivation for allowing access to all these lower level pieces of information is to let researchers to attempt novel, unusual experiments. The space of multi-agent RL has not yet been comprehensively explored, and there are many perfectly plausible reasons you might want access to other agents rewards, observations, and so on. For an API to be universal in an emerging field, it inherently has to allow access to all the information researchers could plausibly want. For this reason we allow access to a fairly straightforward set of lower level attributes and methods in addition to the standard higher level API. As we outline in Section 6.5, we've structured PettingZoo in a way such that including these low-level features doesn't introduce engineering overhead in creating environments, as discussed further in the documentation website.

To handle environments where different agents can be present on each reset of an environment, PettingZoo has an optional `possible_agents` attribute which lists all the agents that might exist in an environment at any point. Environments which generate arbitrary numbers or types of agents will not define a `possible_agents` list, requiring the user to check for new agents being instantiated as the environment runs. After resetting the environment, the `agents` attribute becomes accessible and lists all agents that are currently active. For similar reasons, `num_agents`, `rewards`, `dones`, `infos`, and `agent_selection` are not available until after a reset.

To handle cases where environments need to have environment agents as per the formal AEC Games model, the standard is to put it into the `agents` with the name `env` and have it take `None` as it's action. We do not require this for all environments by default as it's rarely used and makes the API more cumbersome, but this is an important feature for certain edge cases in research. This connects to the formal model in that, when this feature is not used, the environment actor from the formal model and the agent actor that acted before it are merged together.

## 6.5  Environment Creation and the Parallel API

PettingZoo environments actually only expose the `reset`, `seed`, `step`, `observe`, `render`, and `close` base methods and the `agents`, `rewards`, `dones`, `infos`, `state` and `agent_iter` base attributes. These are then wrapped to add the `last` method. Only having environments implement primitive methods makes creating new environments simpler, and reduces code duplication. This has the useful side effect of allowing all PettingZoo environments to be easily changed to an alternative API by simply writing a new wrapper. We've actually already done this for the default environments and added an additional "parallel API" to them that's almost identical to the RLlib POSG-based API via a wrapper. We added this secondary API because in environments with very large numbers of agents, this can improve runtime by reducing the number of Python function calls.

# 7  Default Environments

Similar to Gym's default environments, PettingZoo includes 63 environments. Half of the included environment classes (MPE, MAgent, and SISL), despite their popularity, existed as unmaintained "research grade" code, have not been available for installation via pip, and have required large amounts of maintenance to run at all before our cleanup and maintainership. We additionally included multiplayer Atari games from Terry and Black [2020], Butterfly environments which are original and

of our own creation, and popular classic board and card game environments. All default environments included are surveyed in depth in Appendix B.

## 8 Adoption

In it's relatively short lifespan, PettingZoo has already achieved a meaningful amount of adoption. It is supported by the following learning libraries: The Autonomous Learning Library [Nota, 2020], AI-Traineree [Laszuk, 2020], PyMARL (ongoing) [Samvelyan et al., 2019], RLlib [Liang et al., 2018], Stable Baselines 2 [Hill et al., 2018] and Stable Baselines 3 [Raffin et al., 2019], similar libraries such as CleanRL [Huang et al., 2020] (through SuperSuit [Terry et al., 2020b]), and Tianshou (ongoing) [Weng et al., 2020]. Perhaps more significantly than any of this, PettingZoo is already being used to teach in both graduate and undergraduate reinforcement learning classes all over the world.

## 9 Conclusion

This paper introduces PettingZoo, a Python library of many diverse multi-agent reinforcement learning environments under one simple API, akin to a multi-agent version of OpenAI's Gym library, and introduces the agent environment cycle game model of multi-agent games.

Given the importance of multi-agent reinforcement learning, we believe that PettingZoo is capable of democratizing the field similar to what Gym previously did for single agent reinforcement learning, making it accessible to university scale research and to non-experts. As evidenced by it's early adoption into numerous MARL libraries and courses, PettingZoo is moving in the direction of accomplishing this goal.

We're aware of one notable limitation of the PettingZoo API. Games with significantly more than 10,000 agents (or potential agents) will have meaningful performance issues because you have to step each agent at once. Efficiently updating environments like this, and inferencing with the associated policies, requires true parallel support which almost certainly should be done in a language other than Python. Because of this, we view this as a practically acceptable limitation.

We see three directions for future work. The first is additions of more interesting environments under our API (possibly from the community, as has happened with Gym). The second direction we envision is a service to allow different researchers' agents to play against each other in competitive games, leveraging the standardized API and environment set. Finally, we envision the development of procedurally generated multi-agent environments to test how well methods generalize, akin to the Gym procgen environments [Cobbe et al., 2019].

## Acknowledgements

Justin Terry was supported during part of this work by the QinetiQ Fundamental Machine Learning Fellowship. Thank you to Kyle Sang for their contributions to the documentation website. Thank you Rohan Potdar and Sang Hyun Son for their contributions to the Butterfly benchmarks. Thank you to Deepthi Raghunandan and Christian Clauss for their contributions to testing and continuous integration. Thank you to the PettingZoo community for the numerous bug reports and contributions to the package, especially Ross Allen and their group.

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
