



(a) The initial setup with two agents and two river tiles. When the river tiles become dirty, they are shown as a brownish color instead.

(b) The result of both agents perform the "clean" action. Both river tiles can be are cleaned since Agent 1's action is resolved first.

Figure 7: Cleanup, a Sequential Social Dilemma Game from Vinitsky et al. [2019].

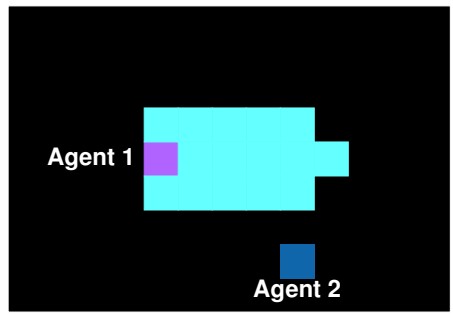

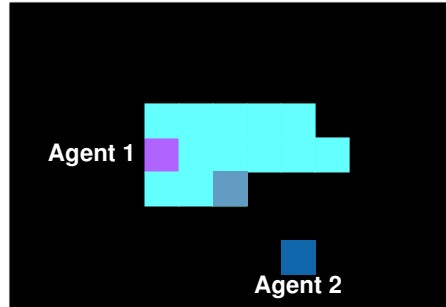

(a) If there are no dirty river tiles in the path of the cleaning beams, the beams will extend to the full length of five tiles.

(b) If there is a dirty river tile in the path of a beam, the beam will stop at the tile, changing it to a "clean" river tile.

Figure 8: An example of Agent 1 using the "clean" action while facing East. The beams extend to a length of up to five tiles. The "main" beam extends directly in front of the agent, while two auxiliary beams start at the tiles directly next to the agent (one to the left and one to the right) and also extend up to five tiles. A beam stops when it hits a dirty river tile.

## A Additional Case Study Information

### A.1 Race Conditions in Sequential Social Dilemma Games

The Sequential Social Dilemma Games, introduced in Leibo et al. [2017], are a kind of MARL environment where good short-term strategies for single agents lead to bad long-term results for all of the agents. New SSD environments, including the *Cleanup* environment, were introduced in Hughes et al. [2018]. All of these have open source implementations in [Vinitsky et al., 2019]. The states of these games are represented by a grid of tiles, where each tile represents either an agent or a piece of the environment. In the *Cleanup* environment, the environment tiles can be empty tiles, river tiles, and apple tiles. Collecting apple tiles results in a reward for the agent and the agents must clean the river tiles with a "cleaning beam" for apple tiles to spawn. The cleaning beam extends in front of agents, one tile at a time, until it hits a dirty river tile ("waste") or extends to its maximum length of 5 tiles. Additionally, two more beams extend in front of the agent—one starting in the tile directly to the agent's left, and one from the tile on the right—until each hits a "waste" tile or reaches a length of 5 tiles. The cleaning beam is shown in Figure 8a. Note that while beams stop at "waste" tiles, they will continue to extend past clean river tiles.

The agents act sequentially in the same order every turn, including the firing of their beams. In the case of two agents trying to occupy the same space, one is chosen randomly, however the tie breaking with regards to the beams is biased, due to a bug. Consider the setup in Figure 7 where each agent chooses the "clean" action for the next step. This results in Agent 1 firing their cleaning beam first, clearing the close river tile. Next, Agent 2 fires their cleaning beam and they are able to clean the

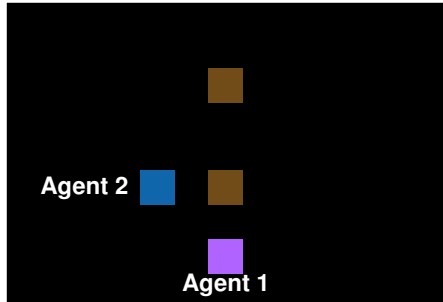

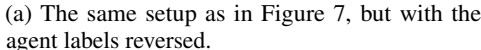

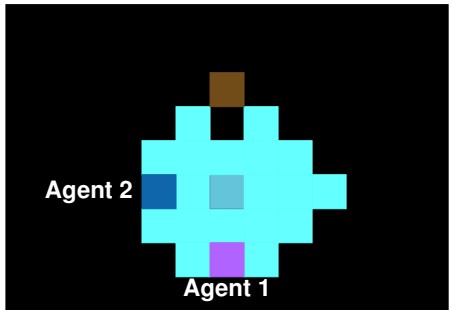

(a) The same setup as in Figure 7, but with the agent labels reversed.

(b) The result of both agents performing the "clean" action, with this agent assignment.

Figure 9: The impact of switching the internal agent order on how the environment evolves. When both agents clean, agent 1's action is resolved first, and the main beam stops when it hits the near dirty river tile, so the far river tile is not cleaned. In Figure 7, Agent 2's beam was able to reach the far beam because Agent 1's beam cleaned the near tile first.

far river tile because the close tile has already been cleared by Agent 1. However, if we keep the same placement and actions but switch the labels of the agents, we get a different result, seen in Figure 9. Now, Agent 1 fires first and hits the close river tile and can no longer reach the far river tile. In situations like these, the observation the second agent's policy is using to act on is going to be inherently wrong, and if it had the true environment state before acting it would very likely wish to make a different choice.

This is a serious class of bug that's very easy to introduce when using parallel action-based APIs, while using AEC games-based APIs prevents the class entirely. In this specific instance, the bug had gone unnoticed for years.

## A.2 Reward Defects in Pursuit

We validated the impact of reward pruning experimentally by training parameter shared Ape-X DQN [Horgan et al., 2018] (the best performing model on pursuit [Terry et al., 2020d]) four times using RLLib [Liang et al., 2017] with and without reward pruning, achieving better results with reward pruning every time and 22.03% more total reward on average Figure 10a, while PPO [Schulman et al., 2017] learned 16.12% more reward on average with this Figure 10b. Saved training logs and all code needed to reproduce the experiments and plots is available in the supplemental materials.

## B Default Environments

This section surveys all the environments that are included in PettingZoo by default.

### Atari

Atari games represent the single most popular and iconic class of benchmarks in reinforcement learning. Recently, a multi-agent fork of the Arcade Learning Environment was created that allows programmatic control and reward collection of Atari's iconic multi-player games [Terry and Black, 2020]. As in the single player Atari environments, the observation is the rendered frame of the game, which is shared between all agents, so there is no partial observability. Most of these games have competitive or mixed reward structures, making them suitable for general study of adversarial and mixed reinforcement learning. In particular, Terry and Black [2020] categorizes the games into 7 different types: 1v1 tournament games, mixed sum survival games (*Space Invaders*, shown in Figure 11a. is an example of this), competitive racing games, long term strategy games, 2v2 tournament games, a four-player free-for-all game and a cooperative game.

### Butterfly

Of all the default environments included, the majority of them are competitive. We wanted to supplement this with a set of interesting graphical cooperative environments. *Pistonball*, depicted

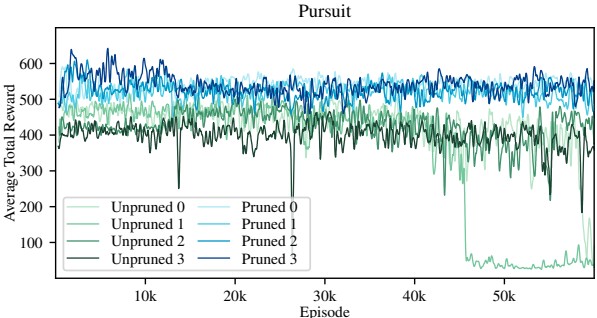

(a) Learning on the *pursuit* environment with and without pruned rewards, using parameter sharing based on Ape-X DQN. This shows an average of an 22.03% improvement by using this method.

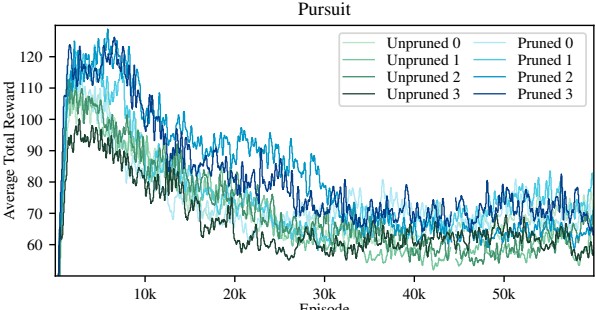

(b) Learning on the *pursuit* environment with and without reward pruning, using parameter sharing based on PPO. Reward pruning increased the total reward by 16.12% on average.

in Figure 11b, is an environment where pistons need to coordinate to move a ball to the left, while only being able to observe a local part of the screen. It requires learning nontrivial emergent behavior and indirect communication to perform well. *Knights Archers Zombies* is a game in which agents work together to defeat approaching zombies before they can reach the agents. It is designed to be a fast paced, graphically interesting combat game with partial observability and heterogeneous agents, where achieving good performance requires extraordinarily high levels of agent coordination. In *Cooperative pong* two dissimilar paddles work together to keep a ball in play as long as possible. It was intended to be a be very simple cooperative continuous control-type task, with heterogeneous agents. *Prison* was designed to be the simplest possible game in MARL, and to be used as a debugging tool. In this environment, no agent has any interaction with the others, and each agent simply receives a reward of 1 when it paces from one end of its prison cell to the other. *Prospector* was created to be a very challenging game for conventional methods—it has two classes of agents with different goals, action spaces, and observation spaces (something many current cooperative MARL algorithms struggle with), and has very sparse rewards (something all RL algorithms struggle with). It is intended to be a very difficult benchmark for MARL, in the same vein as Montezuma's Revenge.

#### Classic

Classical board and card games have long been some of the most popular environments in reinforcement learning [Tesauro, 1995, Silver et al., 2016, Bard et al., 2019]. We include all of the standard multiplayer games in RLCard [Zha et al., 2019]: *Dou Dizhu*, *Gin Rummy*, *Leduc Hold'em*, *Limit Texas Hold'em*, *Mahjong*, *No-limit Texas Hold'em*, and *Uno*. We additionally include all AlphaZero games, using the same observation and action spaces—*Chess* and *Go*. We finally included *Backgammon*, *Connect Four*, *Checkers*, *Rock Paper Scissors*, *Rock Paper Scissors Lizard Spock*, and *Tic Tac Toe* to add a diverse set of simple, popular games to allow for more robust benchmarking of RL methods.

#### MAgent

The MAgent library, from Zheng et al. [2017] was introduced as a configurable and scalable environment that could support thousands of interactive agents. These environments have mostly been studied as a setting for emergent behavior [Pokle, 2018], heterogeneous agents [Subramanian et al., 2020], and efficient learning methods with many agents [Chen et al., 2019]. We include a

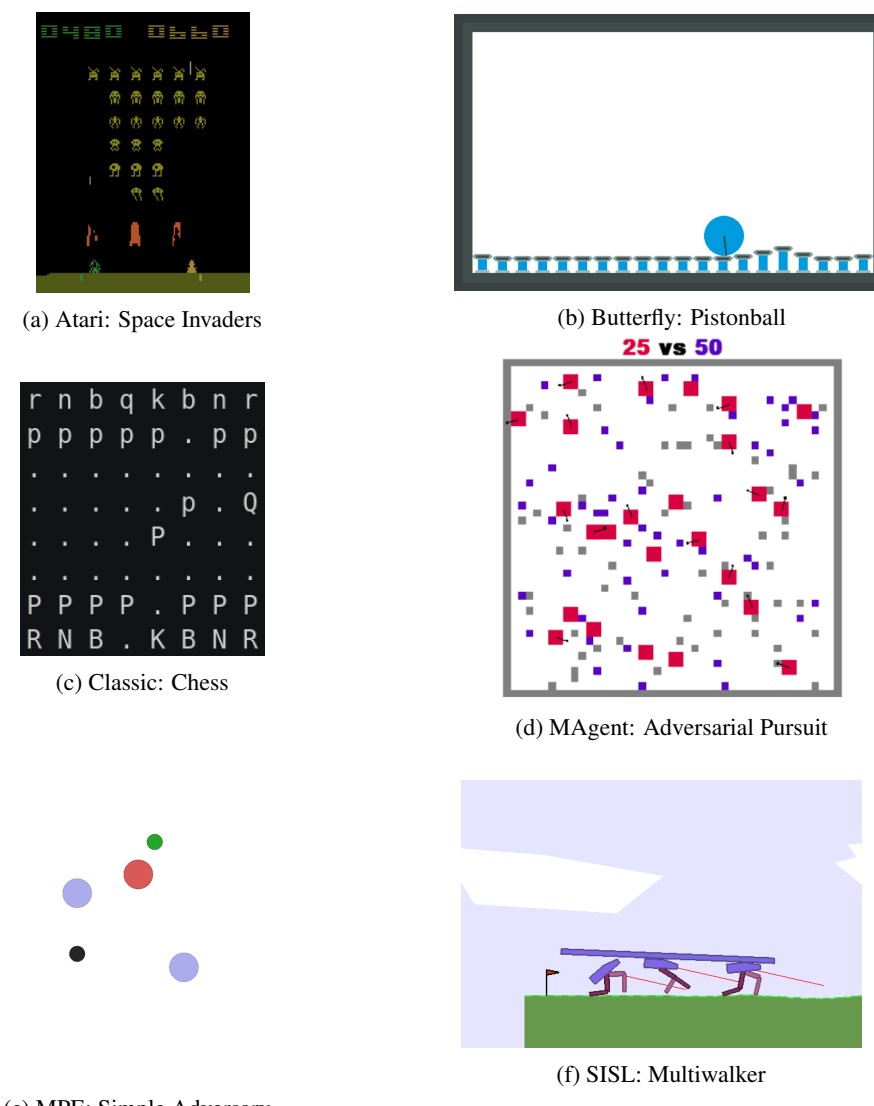

(a) Atari: Space Invaders

(b) Butterfly: Pistonball

(c) Classic: Chess

(d) MAgent: Adversarial Pursuit

(e) MPE: Simple Adversary

(f) SISL: Multiwalker

Figure 11: Example environments from each class.

number of preset configurations, for example the *Adversarial Pursuit* environment shown in Figure 11d. We make a few changes to the preset configurations used in the original MAgent paper. The global "minimap" observations in the battle environment are turned off by default, requiring implicit communication between the agents for complex emergent behavior to occur. The rewards in *Gather* and *Tiger-Deer* are also slightly changed to prevent emergent behavior from being a direct result of the reward structure.

**MPE**

The Multi-Agent Particle Environments (MPE) were introduced as part of Mordatch and Abbeel [2017] and first released as part of Lowe et al. [2017]. These are 9 communication oriented environments where particle agents can (sometimes) move, communicate, see each other, push each other around, and interact with fixed landmarks. Environments are cooperative, competitive, or require team play. They have been popular in research for general MARL methods Lowe et al. [2017], emergent communication [Mordatch and Abbeel, 2017], team play [Palmer, 2020], and much more. As part of their inclusion in PettingZoo, we converted the action spaces to a discrete space which is the Cartesian product of the movement and communication action possibilities. We also added

comprehensive documentation, parameterized any local reward shaping (with the default setting being the same as in Lowe et al. [2017]), and made a single render window which captures all the activities of all agents (including communication), making it easier to visualize.

**SISL**

We finally included the three cooperative environments introduced in Gupta et al. [2017]: *Pursuit*, *Waterworld*, and *Multiwalker*. *Pursuit* is a standard pursuit-evasion game Vidal et al. [2002] where pursuers are controlled in a randomly generated map. Pursuer agents are rewarded for capturing randomly generated evaders by surrounding them on all sides. *Waterworld* is a continuous control game where the pursuing agents cooperatively hunt down food targets while trying to avoid poison targets. *Multiwalker* (Figure 11f) is a more challenging continuous control task that is based on Gym's *BipedalWalker* environment. In *Multiwalker*, a package is placed on three independently controlled robot legs. Each robot is given a small positive reward for every unit of forward horizontal movement of the package, while they receive a large penalty for dropping the package.

### B.1 Butterfly Baselines

Whne environments are introduced to the literature, it is customary for them to include baselines to provide a general sense of the difficulty of the environment and to provide something to compare against. We do this here for the Butterfly environments that this library introduces for the first time; similar baselines exist in the papers introducing all other environments. For our baseline learning method we used used fully parameter shared PPO [Schulman et al., 2017] from Stable-Baselines3 (SB3) [Raffin et al., 2019]. We use the SuperSuit wrapper library [Terry et al., 2020c] for preprocessing similar to that in Mnih et al. [2015], convert the observations to grayscale, resize them to 96x96 images, and use frame-stacking to combine the last four observations. Furthermore, for cooperative_pong_v3 and knights_archers_zombies_v7, we invert the color of alternating agent's observations by subtracting it from the maximum observable value to improve learning and differentiate which agent type an observation came from for the parameter shared neural network, per Terry et al. [2020a]. On the prospector_v4 environment, we add an extra channel to the observations which is set to the maximum possible value if the agent belongs to the opposite agent type, else zero. Both these modifications allow us to use parameter-shared PPO across non-homogeneous agents. On prospector_v4 we also pad observation and agent spaces as described in Terry et al. [2020a] to allow for learning with a single fully parameter shared neural network.

After tuning hyperparameters with RL Baselines3 Zoo [Raffin, 2020], our baselines learns an optimal policy in the Pistonball environment and Cooperative Pong environments and learns reasonably in the Knights Archers Zombies and Prospector environments without achieving optimal policies. Plots showing results of 10 training runs of the best hyperparameters are shown in Figure 12. All code and hyperparameters for these runs is available at `https://github.com/jkterry1/Butterfly-Baselines`.

## C  Formal Definitions

### C.1  Partially Observable Stochastic Games

The formal definition of a POSG is shown in Definition 1. This definition can be viewed as the typical Stochastic Games model [Shapley, 1953] with the addition of POMDP-style partial observability.

**Definition 1.** A *Partially-Observable Stochastic Game* (POSG) is a tuple $\langle \mathcal{S}, s_0, N, (\mathcal{A}_i)_{i \in [N]}, P, (R_i)_{i \in [N]}, , (\Omega_i)_{i \in [N]}, , (O_i)_{i \in [N]} \rangle$, where:

- $\mathcal{S}$ is the set of possible *states*.

- $s_0$ is the *initial state*.

- $N$ is the *number of agents*. The *set of agents* is $[N]$.

- $\mathcal{A}_i$ is the set of possible *actions* for agent $i$.

- $P \colon \mathcal{S} \times \prod_{i \in [N]} \mathcal{A}_i \times \mathcal{S} \to [0, 1]$ is the *transition function*. It has the property that for all $s \in \mathcal{S}$, for all $(a_1, a_2, \ldots, a_N) \in \prod_{i \in [N]} \mathcal{A}_i$, $\sum_{s' \in \mathcal{S}} P(s, a_1, a_2, \ldots, a_N, s') = 1$.

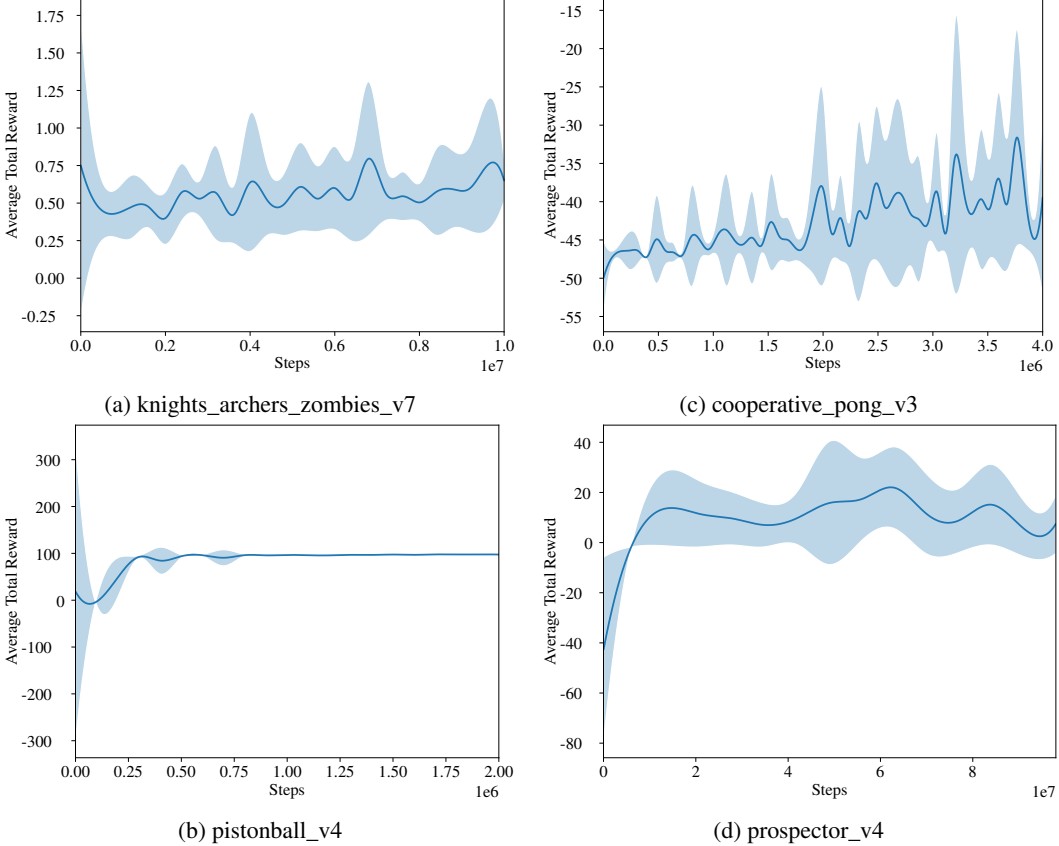

(a) knights_archers_zombies_v7

(b) pistonball_v4

(c) cooperative_pong_v3

(d) prospector_v4

Figure 12: Total reward when learning on each Butterfly environment via parameter-shared PPO.

- $R_i \colon \mathcal{S} \times \prod_{i \in [N]} \mathcal{A}_i \times \mathcal{S} \to \mathbb{R}$ is the *reward function* for agent $i$.

- $\Omega_i$ is the set of possible *observations* for agent $i$.

- $O_i \colon \mathcal{A}_i \times \mathcal{S} \times \Omega_i \to [0,1]$ is the *observation function*. It has the property that $\sum_{\omega \in \Omega_i} O_i(a, s, \omega) = 1$ for all $a \in \mathcal{A}_i$ and $s \in \mathcal{S}$.

### C.2 Extensive Form Games

The definition given here follows closely that of Osborne and Rubinstein [1994], to which we refer the reader for a more in-depth discussion of Extensive Form Games and their formal definition.

**Definition 2.** An Extensive Form Game is defined by:

- A set of agents $[N] = \{1, 2, \ldots, N\}$.

- A "Nature" player denoted as "agent" 0. For convenience, we define $\mathcal{N} := [N] \cup \{0\}$. The Nature agent is responsible for describing the random, stochastic, or luck-based elements of the game, as described below.

- A set $\tilde{\mathcal{A}}$ of *action sequences*. An action sequence is a tuple $\vec{a} = (a_1, a_2, \ldots, a_k)$ where each element indicates an action taken by an agent. In infinite games, action sequences need not be finite. The set $\tilde{\mathcal{A}}$ indicates all possible sequences of actions that may be taken in the game (i.e., "histories" of players' moves or agents' actions). It satisfies the following properties:

    - The empty sequence is in the set: $\varnothing \in \tilde{\mathcal{A}}$.
    - If $(a_1, \ldots, a_k) \in \tilde{\mathcal{A}}$, then for $l < k$ we also have $(a_1, \ldots, a_l) \in \tilde{\mathcal{A}}$.

- In infinite games, if an infinite sequence $(a_1, a_2, \dots)$ satisfies the property that for all $k$, $(a_1, a_2, \dots, a_k) \in \tilde{\mathcal{A}}$, then $(a_1, a_2, \dots) \in \tilde{\mathcal{A}}$.

For a finite sequence $\vec{a} = (a_1, \dots, a_k)$, denote by $(\vec{a}, a)$ the sequence $(a_1, \dots, a_k, a)$. Then the set of actions available in the next turn following a sequence $\vec{a}$ is given by $\mathcal{A}(\vec{a}) := \{a \mid (\vec{a}, a) \in \tilde{\mathcal{A}}\}$ (for convenience, we define $\mathcal{A}(\vec{a}) = \varnothing$ if $\vec{a}$ is infinite). We say a sequence of actions $\vec{a}$ is *terminal* if it is either infinite or if it is a maximal finite sequence, i.e. $\vec{a}$ is terminal if and only if $\mathcal{A}(\vec{a}) = \varnothing$. We denote the set of terminal sequences by $T := \{\vec{a} \mid \mathcal{A}(\vec{a}) = \varnothing\}$.

- A function $\tau \colon (\tilde{\mathcal{A}} \setminus T) \to \mathcal{N}$, which specifies the agent whose turn it is to act next after a given sequence of actions. Note that this is not stochastic, but random player order can be captured by inserting a Nature turn.

- A probability distribution $P(\vec{a}, \cdot)$ for Nature's actions. It is defined only for action sequences for which Nature acts next, i.e. sequences $\vec{a} \in \tilde{\mathcal{A}}$ for which $\tau(\vec{a}) = 0$. Specifically, $P(\vec{a}, a)$ is the probability that Nature takes action $a$ after the sequence of actions $\vec{a}$ has occurred.

- For each agent $i \in [N]$, a *partition* $H_i$ of the sequences of actions $\tilde{\mathcal{A}}_i := \{\vec{a} \mid \tau(\vec{a}) = i\}$. The partition $H_i$ is called the *information partition* of agent $i$, and elements of $H_i$ are called *information sets*. For convenience, define $H := \bigcup_{i \in [N]} H_i$. The information sets must obey the additional property that for any information set $h \in H$ and any two action sequences $\vec{a}, \vec{a}' \in H$, we have $\tau(\vec{a}) = \tau(\vec{a}')$ and $\mathcal{A}(\vec{a}) = \mathcal{A}(\vec{a}')$.

- For each agent $i \in [N]$, a *reward function* $R_i \colon T \to \mathbb{R}$.

## C.3 Agent Environment Cycle Games

As mentioned in Section 5, the stochastic nature of the state transitions is modeled as an "environment" agent, which does not take an action but rather transitions randomly from the current state to a new state according to some given probability distribution. With the stochasticity of state transitions separated out as a distinct "environment" agent, we can then model the transitions of the actual agents deterministically. To this end, each (non-environment) agent $i$ has a deterministic transition function $T_i$ which depends only on the current state and the action taken, while the environment has a stochastic transition function $P$ which transitions to a new state randomly depending on the current state (it may depend on the actions taken previously by the agents, since the current state is determined by these actions).

**Definition 3.** An *Agent-Environment Cycle Game* (AEC Game) is a tuple $\langle \mathcal{S}, s_0, N, (\mathcal{A}_i)_{i \in [N]}, (T_i)_{i \in [N]}, P, (\mathcal{R}_i)_{i \in [N]}, (R_i)_{i \in [N]}, , (\Omega_i)_{i \in [N]}, , (O_i)_{i \in [N]}, , \nu \rangle$, where:

- $\mathcal{S}$ is the set of possible *states*.

- $s_0$ is the *initial state*.

- $N$ is the *number of agents*. The agents are numbered $1$ through $N$. There is also an additional "environment" agent, denoted as agent $0$. We denote the set of agents along with the environment by $\mathcal{N} := [N] \cup \{0\}$.

- $\mathcal{A}_i$ is the set of possible *actions* for agent $i$. For convenience, we further define $\mathcal{A}_0 = \{\varnothing\}$ (i.e., a single "null action" for environment steps) and $\mathcal{A} := \bigcup_{i \in \mathcal{N}} \mathcal{A}_i$.

- $T_i \colon \mathcal{S} \times \mathcal{A}_i \to \mathcal{S}$ is the *transition function for agents*. State transitions for agent actions are deterministic.

- $P \colon \mathcal{S} \times \mathcal{S} \to [0, 1]$ is the *transition function for the environment*. State transitions for environment steps are stochastic: $P(s, s')$ is the probability that the environment transitions into state $s'$ from state $s$.

- $\mathcal{R}_i \subseteq \mathbb{R}$ is the set of possible rewards for agent $i$. We assume this is *finite*.

- $R_i \colon \mathcal{S} \times \mathcal{N} \times \mathcal{A} \times \mathcal{S} \times \mathcal{R}_i \to [0, 1]$ is the *reward function* for agent $i$. $\mathcal{R}_i \subseteq \mathbb{R}$ denotes the set of all possible rewards for agent $i$ (which we assume to be finite).

$R_i$ is the *reward function* for agent $i$. The set of all possible rewards for each agent is assumed to be finite, which we denote $\mathcal{R}_i \subseteq \mathbb{R}$. It is *stochastic*: $R_i(s, j, a, s', r)$ is the probability of agent $i$ receiving reward $r$ when agent $j$ takes action $a$ while in state $s$, and the game transitions to state $s'$. We also define $\mathcal{R} := \bigcup_{i \in [N]} \mathcal{R}_i$.

- $\Omega_i$ is the set of possible *observations* for agent $i$.

- $O_i \colon \mathcal{S} \times \Omega_i \to [0, 1]$ is the *observation function* for agent $i$. $O_i(s, \omega)$ is the probability of agent $i$ observing $\omega$ while in state $s$.

- $\nu \colon \mathcal{S} \times \mathcal{N} \times \mathcal{A} \times \mathcal{N} \to [0, 1]$ is the *next agent* function. This means that $\nu(s, i, a, j)$ is the probability that agent $j$ will be the next agent permitted to act given that agent $i$ has just taken action $a$ in state $s$. This should attribute a non-zero probability only when $a \in \mathcal{A}_i$.

In this definition, the game starts in state $s_0$ and the environment agent acts first. Having the environment agent act first allows the first actual agent to act to be determined randomly if desired (choosing the first agent deterministically can be done easily by having the environment simply do nothing in this first step). The game then evolves in "turns" where in each turn an agent $i$ receives an observation $\omega_i \in \Omega_i$ (any given observation $\omega$ is seen with probability $O_i(s, \omega)$) and, based on this observation, chooses an action $a_i \in \mathcal{A}_i$. The game then transitions from the current state $s$ to a new state $s'$ according to the transition function. If $i \in [N]$, the state transition is deterministically $T_i(s, a_i)$. If $i = 0$, the new state is stochastic, so state $s'$ occurs with probability $P(s, s')$. Then, a new agent $i'$ is determined according to the "next agent" function, so that $i'$ is next to act with probability $\nu(s, i, a_i, i')$. The observation $\omega_i$ that is received is random, occurring with probability $O_i(s, \omega_i)$. Note that we can allow for the state to transition randomly in response to an agent's action by simply inserting an "environment step" immediately following an agent's action, by setting $\nu(s, i, a_i, 0) = 1$ and allowing the following environment step to transition the state randomly. At every step, every agent $j$ receives the partial reward $r'$ with probability $R_j(s, i, a_i, s', r')$.

# D   Omitted Proofs

## D.1   POSGs are Equivalent to AEC Games

The inclusion of the stochastic $\nu$ (next-agent) function in the definition of AEC games allows for capturing many turn-based games with complex turn orders (consider Uno, for instance, where players may be skipped or the order reversed). It is not immediately obvious that this allows for representing games in which agents act simultaneously. However, we show here that in fact AEC games can be used to theoretically model games with simultaneous actions.

To see this, imagine simulating a POSG by way of a "black box" which takes the actions of all agents simultaneously, and then — one by one — feeds them to a purpose-built AEC game whose states are designed to "encode" each agent's action, "queueing" them up over the course of $N$ steps (one for each agent). Once all of the actions have been fed to the AEC game, a single environment step resolves these "queued up" actions all at once. If we design the AEC game in the right way, this total of $N + 1$ steps ($N$ for queueing the actions, and one for the environment to resolve the joint action) produces an outcome that is identical to the result of a single step in the original POSG. This is formalized below.

**Theorem 1.** *For every POSG, there is an equivalent AEC Game.*

*Proof of Theorem 1.* Let $G = \langle \mathcal{S}, N, \{\mathcal{A}_i\}, P, \{R_i\}, \{\Omega_i\}, \{O_i\} \rangle$ be a POSG. To prove this, it will be necessary to show precisely what is meant by "equivalent." We will construct a new AEC Game $G_{\mathrm{AEC}}$ in such a way that for every $N + 1$ steps of $G_{\mathrm{AEC}}$ the probability distribution over possible states is identical to the state distribution for $G$ after a single step, the distributions over observations received by each agent is identical in $G$ and in $G_{\mathrm{AEC}}$, and the reward obtained by each agent is the same.

We define $G_{\mathrm{AEC}}$ as follows:

$$G_{\mathrm{AEC}} = \langle \mathcal{S}', N, \{\mathcal{A}_i\}, \{T_i\}, P', \{R_i'\}, \{\Omega_i\}, \{O_i'\}, \nu \rangle$$

where

- $\mathcal{S}' = \mathcal{S} \times \mathcal{A}_1 \times \mathcal{A}_2 \times \cdots \times \mathcal{A}_N$. That is, an element of $\mathcal{S}'$ is a tuple $(s, a_1, a_2, \ldots, a_N)$ where $s \in \mathcal{S}$ and for each $i \in [N]$, $a_i \in \mathcal{A}_i$.

- $T_i((s, a_1, a_2, \ldots, a_i, \ldots, a_N), a_i') = (s, a_1, a_2, \ldots, a_i', \ldots, a_N)$.

- For $\mathbf{s} = (s, a_1, a_2, \ldots, a_N)$ and $\mathbf{s}' = (s', a_1, a_2, \ldots, a_N)$, we define $P'(\mathbf{s}, \mathbf{s}') = P(s, a_1, a_2, \ldots, a_N, s')$. If $\mathbf{s}$ and $\mathbf{s}'$ are such that $a_i \neq a_i'$ for any $i \in [N]$, then $P'(\mathbf{s}, \mathbf{s}') = 0$.

- For $\mathbf{s} = (s, a_1, a_2, \ldots, a_N)$, $\mathbf{s}' = (s', a_1, a_2, \ldots, a_N)$, and $\mathbf{r} = R_i(s, a_1, a_2, \ldots, a_N, s')$, we let $R_i'(\mathbf{s}, 0, \varnothing, \mathbf{s}', \mathbf{r}) = 1$. We define $R_i' = 0$ for all other cases.

- $O_i'(s, a_1, a_2, \ldots, a_N) = O_i(s)$

- $\nu((s, a_1, a_2, \ldots, a_N), i, a_i', j) = 1$ if $j \equiv i + 1 \pmod{N+1}$ (and equals 0 otherwise).

The AEC game $G_{\text{AEC}}$ begins with agent 1. If the initial state of the POSG $G$ was $s_0$, then the initial state of $G_{\text{AEC}}$ is $(s_0, \cdot, \cdot, \ldots, \cdot)$, where all but the first element of the tuple are chosen arbitrarily.

Let $P_{t,s}$ be the probability that the POSG $G$ is in state $s$ after $t$ steps. For an action vector $\mathbf{a} = (a_1, \ldots, a_N) \in \mathcal{A}_1 \times \cdots \times \mathcal{A}_N$, let $P_{t,s,\mathbf{a}}'$ be the probability that $G_{\text{AEC}}$ is in state $(s, a_1, \ldots, a_N)$ after $t$ steps. Finally, let $P_{t,s}' = \sum_{\mathbf{a} \in \mathcal{A}_1 \times \cdots \times \mathcal{A}_N} P_{t,s,\mathbf{a}}'$.

Trivially, $P_{0,s} = P_{0,s}'$ for all $s \in \mathcal{S}$. Now, suppose that after $t$ steps of $G$, $P_{t,s} = P_{t(N+1),s}'$ for all $s \in \mathcal{S}$ (our inductive hypothesis). For any joint action $\mathbf{a} = (a_1, \ldots, a_N)$, the state distribution of $G$ at step $t + 1$ if the joint action $\mathbf{a}$ is taken is given by $P_{t+1,s'} = P_{t,s} \cdot P(s, a_1, \ldots, a_N, s')$. Further, the reward obtained by agent $i$ for this joint action, if the new state is $s'$, is $R_i(s, a_1, \ldots, a_N, s')$. Let $\mathbf{s} = (s, a_1, \ldots, a_N)$ and $\mathbf{s}' = (s', a_1, \ldots, a_N)$. Then, in $G_{\text{AEC}}$, if the agents take actions $a_1, a_2, \ldots, a_N$ respectively on their turns, the state distribution of $G_{\text{AEC}}$ at step $(t + 1)(N + 1)$ is given by $P_{(t+1)(N+1),s'}' = P_{(t+1)(N+1),s',\mathbf{a}}' = P_{t(N+1),s}' P'(\mathbf{s}, \mathbf{s}')$. By the inductive hypothesis, $P_{t(N+1),s}' = P_{t,s}$, and by the definition of $P'(\mathbf{s}, \mathbf{s}')$ in $G_{\text{AEC}}$, it is clear that $P'(\mathbf{s}, \mathbf{s}') = P(s, a_1, \ldots, a_N, s')$. Thus, $P_{(t+1)(N+1),s'}' = P_{t,s} P(s, a_1, \ldots, a_N, s') = P_{t+1,s'}$.

The above establishes a strict equivalence between the state distributions of $G$ at step $t$ and $G_{\text{AEC}}$ at step $t(N + 1)$ for any $t$. Between steps $t(N + 1) + 1$ and $(t + 1)(N + 1)$ of $G_{\text{AEC}}$, each agent in turn receives an observation and then chooses its action. Specifically, agent $i$ acts at step $t(N) + i$ immediately after receiving an observation $\omega_i$ with probability $O_i'(s, a_1, \ldots, a_N) = O_i(s)$. Thus, the marginal probability distribution (when conditioned on transitioning into state $s$) of the observation received by agent $i$ immediately after acting at time $t$ in $G$ is identical to the marginal distribution of the observation received by $i$ immediately before acting at time $t(N + 1) + i$ in $G_{\text{AEC}}$, i.e. $\Pr_{G,t}(\omega_i = \omega \mid s_t = s) = \Pr_{G_{\text{AEC}},t(N+1)+i}(\omega_i = \omega \mid s_{t(N+1),0} = s)$.

The second part of the equivalence is observing that the reward received by an agent $i$ in $G$ after the joint action $\mathbf{a}$ is taken is equivalent to the total reward received by agent $i$ in $G_{\text{AEC}}$ across all steps from $t(N + 1) + 1$ through $(t + 1)(N + 1)$ when the agents take actions $a_1, \ldots, a_N$ respectively. We can see that this is indeed the case, since the rewards received by agent $i$ in $G_{\text{AEC}}$ from step $t(N + 1) + 1$ through step $(t + 1)(N + 1)$ is 0 at every step but the environment step $(t + 1)(N + 1)$. By definition of $R'$ in $G_{\text{AEC}}$, $R_i'(\mathbf{s}, 0, \varnothing, \mathbf{s}', R_i(s, a_1, \ldots, a_N, s')) = 1$, so the total reward received by any agent $i$ in $G_{\text{AEC}}$ is $R_i(s, a_1, \ldots, a_N, s')$. This establishes the second part of our equivalence (that the reward at step $t(N + 1)$ in $G_{\text{AEC}}$ is identical to the reward at step $t$ of $G$, if the actions are the same). $\qquad\square$

One way to think of this construction is that the actions are still resolved simultaneously via the *environment step* (which is responsible for the stochastic state transition and the production of rewards); we simply break down the production of the joint action into smaller units whereby each agent chooses and "locks in" their actions one step at a time. A toy example to see this equivalence is to imagine a multiplayer card game in which each player has a hand of cards and each turn consists of all players choosing one card from their hand which is revealed simultaneously with all other players. An equivalent game has each player in sequence choosing a card and placing it face down on their turn, followed by a final action (the "environment step" in which all players simultaneously reveal their selected card.

At first, it may appear as though the AEC game is in fact *more* powerful than the POSG, since in addition to being able to handle simultaneous-action games as shown above, it can represent sequential games, including sequential games with complex and dynamic turn orders such as Uno (another aspect of our AEC definition that seems more general than in POSGs is the fact that the reward function in an AEC game is stochastic, allowing rewards to be randomly determined). However, it turns out that a POSG can be used to model a sequential Handling the stochastic rewards and stochastic next-agent function is non-obvious and is omitted here due to space constraints; the construction and proof can be found in Appendix D.1.

We next show how to convert an AEC game to a POSG for the case of deterministic rewards.

**Definition 4.** An AEC Game

$$G = \langle \mathcal{S}, N, \{\mathcal{A}_i\}, \{T_i\}, P, \{R_i\}, \{\Omega_i\}, \{O_i\}, \nu \rangle$$

is said to have *deterministic rewards* if for all $i, j \in \mathcal{N}$, all $a \in \mathcal{A}_j$, and all $s, s' \in \mathcal{S}$, there exists a $R_i^*(s, j, a, s')$ such that $R_i(s, j, a, s', r) = 1$ for $r = R_i^*(s, j, a, s')$ (and 0 for all other $r$).

Notice that an AEC Game with deterministic rewards may still depend on the new state $s'$ which can itself be stochastic in the case of the environment ($j = 0$).

**Theorem 2.** *Every AEC Game with deterministic rewards has an equivalent POSG.*

*Proof.* Suppose we have an AEC game

$$G = \langle \mathcal{S}, N, \{\mathcal{A}_i\}, \{T_i\}, P, \{R_i\}, \{\Omega_i\}, \{O_i\}, \nu \rangle$$

with deterministic rewards. We define $G_{\text{POSG}} = \langle \mathcal{S}', N, \{\mathcal{A}_i\}, P', \{R_i'\}, \{\Omega_i\}, \{O_i\} \rangle$ as follows.

- $\mathcal{S}' = \mathcal{S} \times \mathcal{N}$

- $P'((s, i), a_1, \ldots, a_N, (s', i')) = \nu(s, i, a_i, s', i') \cdot \Pr(s' \mid s, i, a_i)$, where

$$\Pr(s' \mid s, i, a_i) = \begin{cases} 1 & \text{if } i > 0, T(s, a_i) = s' \\ P(s, s') & \text{if } i = 0 \\ 0 & \text{o/w} \end{cases}$$

- $R_i'((s, j), a, (s', j')) = R_i^*(s, j, a, s')$

In this construction, the new state in the POSG encodes information about which agent is meant to act. State transitions in the POSG therefore encode both the state transition of the original AEC game and the transition for determining the next agent to act. In each step, the state transition depends only on the agent who's turn it is to act (which is included as part of the state).

This construction adapts POSGs to be strictly turn-based so that it is able to represent AEC Games. $\square$

We now present the full proof.

**Theorem 3.** *Every AEC Game has an equivalent POSG.*

*Proof.* Suppose we have an AEC game $G = \langle \mathcal{S}, N, \{\mathcal{A}_i\}, \{T_i\}, P, \{R_i\}, \{\Omega_i\}, \{O_i\}, \nu \rangle$, and $\mathcal{R}$ is the (finite) set of all possible rewards. We define $G_{\text{POSG}} = \langle \mathcal{S}', N, \{\mathcal{A}_i\}, P', \{R_i'\}, \{\Omega_i\}, \{O_i\} \rangle$ as follows.

The state set is $\mathcal{S}' = \mathcal{S} \times \mathcal{N} \times \mathcal{R}^N$. An element of $\mathcal{S}'$ is a tuple $(s, i, \mathbf{r})$, where $\mathbf{r} = (r_1, r_2, \ldots, r_N)$ is a vector of rewards for each agent.

The transition function is given by

$$P'((s, i, \mathbf{r}), a_1, a_2, \ldots, a_N, (s', i', \mathbf{r}')) =$$
$$\nu(s, i, a_i, s', i') \Pr(s' \mid s, i, a_i) \prod_{j \in [N]} R_j(s, i, a_i, s', \mathbf{r}'_i)$$

where
$$\Pr(s' \mid s, i, a_i) = \begin{cases} 1 & \text{if } i > 0 \text{ and } T(s, a_i) = s' \\ P(s, s') & \text{if } i = 0 \\ 0 & \text{o/w} \end{cases}$$

The reward function is given by $R'_i((s, j, \mathbf{r}), a, (s', j', \mathbf{r}')) = \mathbf{r}'_i$ $\qquad\qquad\qquad$ □