# OpenReview forum: "PettingZoo: Gym for Multi-Agent Reinforcement Learning"
_NeurIPS.cc/2021/Conference — NeurIPS 2021 Poster_

### Official Review · Reviewer_BcrF · 2021-07-13

**Rating:** 5
**Confidence:** 4

**Summary:**

The paper introduces PettingZoo, a Python library for multi-agent RL tasks. It also discusses the API choices of said library, and how they might favoriably compare to other approaches available.

**Limitations And Societal Impact:**

Some limitations discussed below. No potential negative societal impact seems likely to incur as a result of this submission.

**Main Review:**

The paper is highly readable and relatively clearly written, with only minor mistypes and glitches (examples: a missing "s" in "observation[s]" in Figure 2, a missing "T" in "[T]he" in line 97, a missing "occur" after "typically" in line 185). It presents the present state of the accompanying library, PettingZoo, relatively well. The main contribution of the submission itself, on top of the library presented, is a discussion of "POSG" and "EFG", and the introduction of "AEC". These acronyms are taken to mean different API choices for multi-agent RL environments. Specifically, "POSG" is used for the API model in which a vector of observations and rewards is returned by the task object, one for each agent, which in turn expects a vector of actions. The "EFG" model is used for the API of OpenSpiel, namely the tree-like structure more natural for board games, in which random effects of the environment (such as during a game of Poker or Blackjack) are represented as "chance nodes". The suggested "AEC" API model has the task define an order in which agents get to observe and act. That order is presented as a (potentially infinite) Python iterator object and is meant to represent situations in which the turn order changes during the game, as it would e.g. in the card game Uno.

The paper discusses a specific bug the authors ran into while using the "POSG" model and argues their model would help preventing that type of bug. On the face of it, this would seem like a worthwhile point, given the complexities of RL research at scale, especially in the multi-agent case. However, this reviewer wasn't entirely convinced by the presentation. Indeed, it seems the paper makes some rather strong claims without being able to give a reference as to why they would be true, and the central anecdote around the bug / race-condition discussed seems overly simplifying.

Examples of statements that seem questionable in the generality in which they are stated:

"[popular previous APIs] promote severe bugs that are hard to detect" (from the abstract, line 11): While that may be true in an extremely general sense in that all RL research easily suffers from bugs that could be described that way, the specific example discussed seems insufficient to illustrate this in the particular case the authors must have meant, see below.

"[Multi-agent RL] remains a significant engineering challenge. This is because, unlike single agent reinforcement learning which has OpenAI's Gym, no de facto standard API exists [...] for how agents interface with environments." (line 18): This is clearly an overstatement in both directions. Single-agent RL remains a significant engineering challenge despite OpenAI's useful library, and although PettingZoo sounds like a very useful contribution to the landscape, its authors can hardly believe it will make multi-agent RL trivial or even easy. It may perhaps make it simpler along some dimensions, but even that is probably better argued empirically, and at any rate not the obvious reading of this statement, although the authors would likely support that statement along these lines.

"[We] discovered [...] [other APIs] do not make good conceptual sense [...] and cannot form the basis of APIs that cleanly handle all types of multi-player environments" (line 35): This is a rather strong statement that would require equally strong arguments. However, looking at the list of specific criticisms of previous APIs, we read (lines 72--): (1) Dummy actions are necessary and (2) changing numbers of agents requires learning code to deal with lists "suddenly" changing sizes. For the OpenSpiel API, the criticisms cited are (lines 89--): (1) It's too complex, especially for beginners or for RL researchers to "ubiquitously use it as [their] mental model of games", (2) the formal definition only has rewards at the end of the game -- which is followed by "this is possible to work around" (by changing that formal definition, perhaps?), but "not ideal", and finally (3) the lack of continuous action spaces -- followed by the admission that this, too, "is not inherent in the [OpenSpiel API]". This sauce is just a bit too weak for the strong praise the chef gave it.

Let's now discuss the anecdotal evidence the submission relates around race-conditions in multi-agent RL. An example is given where two agents observe a state, each gets a chance to act on it and they chose to act in incompatible ways. "Tie-breaking" is then happening in the environment, e.g., by one agent being selected to be allowed to perform its action as intended, while the other agent might be denied, or have its action applied to the situation after the first agent got to modify the state with its action. The authors mention a situation where this tie-breaking was happening in a biased way and suggest that serializing the interaction with the environment would get around this issue. As indeed it would. However, the claim that this "is more closed [sic!] models how computer games are executed in code" (line 211) is naive at best. While this reviewer is no expert in game engine development, it seems more likely modern games are implemented around event loops interacting with a main rendering thread. That is, the state of the game might continuously change (units move, wind blows, bombs explode, etc), while the players have a chance to interact with that game in an asynchronous way, perhaps rate-limited to a certain number of actions per minute by either the system or natural constraints in hardware, network latency, or human motor control. When doing RL research on such games, this reality needs to be taken into account. And it is, one way or the other, in results like AlphaStar (which gets around asynchronous environments by trying to be fast enough at test time, and make the environment ignore the occasional action that can no longer be applied, like trying to construct a unit with a building that's already destroyed). The challenges of real-world interactions like robotics or self-driving cars are similar, only worse. Now, it might reasonably be argued that it's too much to ask from an API to cover chess, StarCraft, and Mars robots all at once. And indeed that might be so -- but that's not exactly an argument in favor of a single new standardized API.

None of this is to say that the library the authors developed doesn't have the potential to be extremely valuable to the community, and might well already have great positive impact. It would seem to this reviewer, however, that online documentation of their API and the design choices behind it is a better format (especially in view of potential future changes) than a research paper.

As a side note, the submission's supplementary material is insufficiently anonymized. This reviewer closed the relevant file right after observing this and believes no undue influence on the double-blinded process has occurred.

UPDATE: Changed rating to 5: Marginally below the acceptance threshold based on author's response.

**Time Spent Reviewing:**

5

---

> ### Author Response · Authors · 2021-08-10
> **Response**
>
> Hey,
>
> We greatly appreciate your extraordinarily thorough review, and I think we were actually able to reasonably resolve all your specific concerns below. However, your principal reason for rating this paper reject appears to be your thought that the work is not suitable for a research publication and should instead simply be included in online documentation. Regarding this, NeurIPS specifically solicits papers that research “Infrastructure (e.g., datasets, competitions, implementations, libraries),” and famous papers have been accepted to NeurIPS in this space as well, such as a paper on PyTorch. Our work introduces a new model for interacting with MARL environments, and presents a library which implements this model, along with justifications for all of the design decisions required to implement this game model as an API. Finally, we include case studies which examine the causes of two distinct, major bugs in popular MARL environments, and how our API and game model would have prevented those bugs. As such, our paper clearly addresses topics NeurIPS is soliciting and includes scientific contributions beyond purely technical ones.
>
> Moreover, PettingZoo has become exponentially more used. During the review period, it is has become the third most installed RL library on PyPI (the repository where pip install comes from) after Gym and Ray/RLlib (you can verify this while preserving anonymity at pepy.tech), and the API has already used in dozens of large third party projects. Papers like this for tools that are widely used by the community must be accepted to incentivize further development and in order for the machine learning community to be able to function healthily.
>
> With these things in mind, it’s my understanding that you have the following concerns with our work. These generally are the result of imprecise writing on our part and we appreciate you pointing these out:
>
> 1) "[Multi-agent RL] remains a significant engineering challenge. This is because, unlike single agent reinforcement learning which has OpenAI's Gym, no de facto standard API exists [...] for how agents interface with environments." was clearly incorrect as you pointed out. We changed this to "[Multi-agent RL] remains a significant engineering challenge. /A large part of this/ is because, unlike single agent reinforcement learning which has OpenAI's Gym, no de facto standard API exists [...] for how agents interface with environments." This is a reasonable thing to say-- without a standard API, code that works with one learning suite or environment requires modification to work with another (sometimes across languages). This was a writing error on our part and we do not believe this change weakens the paper. We would also like to clarify that we are only discussing the engineering challenges of MARL, not the difficulty of solving MARL problems. We believe this should be clear from the phrasing of this section.
>
> 2) You state that "'[We] discovered [...] [other APIs] do not make good conceptual sense [...] and cannot form the basis of APIs that cleanly handle all types of multi-player environments' (line 35): This is a rather strong statement that would require equally strong arguments."
>
> 2a) Regarding the ability to cleanly handle all types of multi-player environments: We believe that, as you quote our paper in mentioning, the requirement of dummy actions to represent strictly turn based games with a POSG API means that POSG based APIs cannot be a clean and general API. Dummy actions complicate user code by forcing users to submit additional actions that serve no purpose. Building APIs with the EFG tree model adds further complexity to the API that makes it less convenient to work with, as plainly shown in Figure 3. This was intended to be covered in bullet point 1 of section 2.2 but was not, and we'll revise this in the paper.
>
> 2b) Regarding the “making good conceptual sense” statement, which was intended to be specific to POSG APIs not EFGs, we've clarified our copy of the paper as such (this reduced claim similarly should not meaningfully weaken our paper). While we’ll make sure this is more clearly stated in the bullets you mention, this particular assertion was clearly drawing support from our case studies in our paper. We dive into how our case studies support this in section 4.
>
> 3) Regarding race conditions:
>
> 3a) You state that sequential agent updates are not closer to how computer games work. I disagree. Ultimately on a single CPU thread, execution happens sequentially, and only one agent can update at a time, even if using the event loop structure. At its core agents literally are updated sequentially in computer games. In your specific example of asynchronous actions/observations in an environment, you pose that the natural thing to do would be to use a POSG model where agents all step at once and the model just “has to deal with” the inherent time mismatches. An alternative would be to change individual agent selection to correspond to the currently acting asynchronously selected agent, which would prevent possible mismatches in order. While not always technically viable due to restricted access to environment internals (e.g. Atari ROMS), such an approach would still pose the advantages of allowing these potential race condition problems to be eliminated even in your event loop case. While these problems are not necessarily always severe, it is worth making design decisions to be able to resolve them. Regardless, as we discuss PettingZoo still supports both AEC game and POSG based APIs should users prefer this route.
>
> 3b) You are correct that observation delays make the race condition problem go away. However, environments of note to researchers without observation delays still can exist, such as popular grid worlds or RK4-type physics simulations (without backtracking), and modeling these in a useful manner remains an important challenge for researchers. We will explicitly clarify this note in the paper.
>
> 4) You disagree that our case studies etc. support our assertion that "[popular previous APIs] promote severe bugs that are hard to detect.” and that POSGs based APIs do not make good conceptual sense. We believe, 3a combined with our other case study in 4.1, which you didn’t raise any issue with in your rebuttal, support these claims. To be clear here, the POSG API doesn’t guarantee that bugs will happen and the AEC games API doesn’t guarantee that bugs won’t happen, the AEC game API allows for the prevention of certain classes of bugs that can be hard to find. We’ll similarly make sure the wording to this end is clear. You’re also likely correct in saying that POSG APIs don’t make good conceptual sense in general is a bit strong for what we argue given our cases. We’ll soften this as well to “POSG-based APIs are not conceptually clear in certain cases,” which does not meaningfully diminish our detailed claims about the AEC games model.
>
> -We also fixed all the typos you noted, thank you for pointing those out.

---

> > ### Comment · Reviewer_BcrF · 2021-08-16
> > **Re: Response**
> >
> > Thanks for your detailed response.
> >
> > > Regarding this, NeurIPS specifically solicits papers that research “Infrastructure (e.g., datasets, competitions, implementations, libraries),” and famous papers have been accepted to NeurIPS in this space as well, such as a paper on PyTorch.
> >
> > I fully agree with that and did not want to give any impression to the contrary.
> >
> > > Our work introduces a new model for interacting with MARL environments, and presents a library which implements this model, along with justifications for all of the design decisions required to implement this game model as an API.
> >
> > That is correct, although I found the justification lacking, as per above.
> >
> > > Finally, we include case studies which examine the causes of two distinct, major bugs in popular MARL environments, and how our API and game model would have prevented those bugs. As such, our paper clearly addresses topics NeurIPS is soliciting and includes scientific contributions beyond purely technical ones.
> >
> > As discussed, I might either have misunderstood the nature of these case studies or just remain slightly unconvinced by them.
> >
> > Regarding the change in language you proposed: I'm very happy to have caused that change. I would probably feel that _a large part of this_ is still a rather strong statement without a clear proof, but it's also certainly a big improvement.
> >
> > Regarding the discussions of race conditions:
> >
> > > Ultimately on a single CPU thread, execution happens sequentially, and only one agent can update at a time, even if using the event loop structure. At its core agents literally are updated sequentially in computer games.
> >
> > The first sentence is correct, but that correctness is carried by the "on a single CPU thread" qualifier. That condition will not be true in modern computer games (e.g., StarCraft II, Dota II, etc). It's also not remotely true for robotics, where one could also imagine a multi-agent setup.
> >
> > This is of course true for most board games (although not for some highly interactive card games like some versions of Uno) and the kinds of toy environments specifically written for multi-agent tasks this community focuses on. And it's fine to focus on these, but I would prefer to acknowledge the somewhat limited nature of this application, especially in light of the "we will solve _everything_" approach many researchers have embraced in the field in recent years.
> >
> > That said, the detailed response above and change in language proposed makes me happy to switch to a "5: Marginally below the acceptance threshold" rating.
> >
> > Please understand that this is no verdict on the library itself, which is probably really good and useful -- in fact, I wonder if a publication on its empirical usefulness might be a good endeavour? That could also tackle some of the hard-to-argue questions like the "reasonability" of one API over another.

---

> > > ### Author Response · Authors · 2021-08-25
> > > **Response**
> > >
> > > Hey, thank you for raising your score.
> > >
> > > Regarding the case race condition problem, I believe that your comment "That condition will not be true in modern computer games" is incorrect. Ultimately in any multiplayer game, there is a point where separate, possibly asynchronous player actions must be combined into the game’s global state. If the actions affect the same part of the game state, which is the situation we consider in tie-breaking, then this process cannot be done by two threads without risk of one thread clobbering the changes of the other. At some point in the process, a synchronous update must occur where one agent’s action is processed first. Ideally, this should be done in tie-breaking code, where the potential bias can be handled gracefully. For example, this race condition occurs in the Unity game engine (which according to Google is the engine used by 50% of all games and which has the popular ML Agents library), which has the widely used Unity ML Agents API. We are unable to find citations for this this race condition specifically that do not de-anonymizing ourselves, but the just documentation of how rigid body collisions are handled in Unity implies this issue (see https://docs.unity3d.com/Manual/class-Rigidbody.html and https://docs.unity3d.com/Manual/ContinuousCollisionDetection.html).
> > >
> > > In robotics, or any continuous-time environment, this type of tie breaking is inconsequential because the probability of two events occuring at the exact same time is zero. These real world environments would likely not utilize a library like PettingZoo, but simulated versions of these environments typically discretize time in some way. You are however correct that the AEC games model cannot represent continuous time. This is what I alluded to in 3b (apologize that I wasn't more explicit with my language there), and this a major reason why we included the secondary POSG based API. We'll clarify this explicitly in the paper.
> > >
> > > Regarding empirical usefulness, we seriously considered doing studies on this, and spoke to researchers who specialize in programming language user studies to this end. In essence, the time required for a user to develop a MARL environment or write code to interact with it from scratch is so time intensive that we were advised that there is no productive way to conduct a conventional user study. However, PettingZoo’s user downloads and adoption by numerous major RL libraries also provide an indication of its usefulness as an API. As I previously alluded to, PettingZoo is the most installed multi-agent library in the world where data is available, it's the 3rd most installed RL library on PyPI (which is where everyone installs pip packages from), and a large fraction of the community has switched to our API (e.g. Autonomously Learning Library, RLlib, Mava, Stable Baselines, CleanRL and other similar libraries via SuperSuit and even SMAC) and Tianshou, Unity ML Agents and Neural MMO and others are all in the process of transitioning to PettingZoo. Alternative APIs to PettingZoo exist (they're what people are replacing with PettingZoo's). This is likely far more compelling empirical evidence than any user study could provide (note that some of these things happened while the paper was under review).
> > >
> > > With this in mind, I do think that the case studies are correct and many people who have read them do as well. While I've heard similar comments to yours at the handful of talks I've given on PettingZoo, those who raised these concerns have always been satisfied after some back and forth. If you get the opportunity, could you please elaborate on what precisely you remain unconvinced of regarding our arguments about race conditions?

---

### Official Review · Reviewer_yy96 · 2021-07-16

**Rating:** 5
**Confidence:** 4

**Summary:**

This paper presents the PettingZoo library containing a set of multiagent environments with a Gym-like interface to users.

**Limitations And Societal Impact:**

yes


**Main Review:**

Originality: this paper is not scientifically novel. According to the claim, the main contribution is "be GYM like for multiagent setting".
However, "OpenAI Gym" is an arxiv paper and has been published nowhere (as far as I know, please correct me if i am wrong).
Clarity: the writing of the paper easy to follow and clearn. It is more of an engineering document or tutorial than a scientific paper to me.
Other: I would suggest the authors to expand the revisions to the ICML submission.

**Time Spent Reviewing:**

1

---

> ### Author Response · Authors · 2021-08-03
> **Scientific Contribution**
>
> Thank you for your time in reviewing our paper.
>
> Our paper has two primary contributions: the theoretical AEC games model of multi-agent systems and the PettingZoo library. While you're correct that the PettingZoo work is not a scientific contribution itself (it’s primarily engineering work), the associated AEC games model we introduce is a novel mathematical and conceptual model of multi-agent games which we argue is more useful in many contexts and which we prove is as expressive as the most popular formalism in multi-agent games for MARL. That is work with clear scientific contribution.
>
> Our work also clearly states the significance of the AEC games model in our work. For instance, the first line of our abstract is "This paper introduces the PettingZoo library and the accompanying Agent Environment Cycle (‘AEC’) games model." Moreover, roughly 2/3rds of the lines in the body of this paper were spent on the AEC games model, not PettingZoo.
>
> Our discussion of AEC games includes a new formal definition, proofs and case studies. Because your review did not address this additional scientific contribution, we would respectfully request that you reconsider your review of our work in light of these contributions.
>
> Regarding Gym, you are correct that it is solely published on ArXiv. However, unlike our paper, the Gym paper did not also introduce a variant of the POMDP model, which likely would’ve made it publishable in a scientific venue.

---

### Official Review · Reviewer_KaoH · 2021-07-16

**Rating:** 7
**Confidence:** 3

**Summary:**

This paper introduces a PettingZoo, a python API similar to OpenAI gym for multi-agent reinforcement learning (MARL). The environment has been in use for several years, and has 62 built in environments, is supported by a number of libraries, and is already used by many other groups and students. A key contribution of the paper is explicating the Agent Environment Cycle (AEC) model that powers PettingZoo, and its key distinguishing features in comparison to alternative MARL models.


**Ethical Concerns:**

No ethical concerns.

**Limitations And Societal Impact:**

They could add some boilerplate on societal impacts.

**Main Review:**

## A principled and democratized framework for multi-agent reinforcement learning.

### Strengths
- Excellent library for multi-agent reinforcement learning.
- The description of the AEC framework in comparison to other models is well constructed.
- Number and description of environments and supporting packages is clear and impressive.

### Weaknesses
- Much of the comparison with other game models is handled in the text, and could be diagrammed more clearly, for instanced in Figure 5. A table could also be used to quickly lookup differences in these approaches, with appropriate subheadings (i.e. race conditions).
- Any quantitative measures on users, implemented games or other comparisons with competing frameworks would be helpful, as part of the challenge in comparing PettingZoo with other libraries is that the arguments are somewhat qualitative. A table here could again help.


### Correctness
I believe the claims are correct.

### Clarity
The manuscript is clear and easy to read.

### Relation to prior work
Relationship with other approaches is present and easy to parse.

### Reproducibility
The code is well maintained and a popular repository on github.

### Additional feedback, comments, suggestions for improvement and questions for the authors
I understand the need to distinguish from past work, but the comparisons with POSG and EFL sometimes come across as a bit biased. For instance ‘ POSGs Based APIs Don’t Make Conceptual Sense For Games Implemented In Code’ mainly concerns race conditions.



**Time Spent Reviewing:**

1

---

> ### Author Response · Authors · 2021-08-03
> **Thank You**
>
> We greatly appreciate your feedback. To answer your points:
>
> -We did not include tables as we believe the claims are too nuanced to represent fairly in simple tables
>
> -We are unaware of any conventional qualitative metrics that are well suited to comparing our API to others. This is a general problem in software engineering research. For example, while there is a sort of community consensus amongst software engineers that Rust is safer than C or that certain aspects of JavaScript are unpleasant, precise qualitative metrics for this do not exist, and instead people tend to make arguments such as ours. Common metrics used in lieu of this are typically adoption rates or surveys on community sentiment for certain libraries. The user base of MARL is sufficiently small and specialized that we can't feasibly conduct a sentiment comparison amongst libraries because so few people we've spoken to have extensively used all the major APIs. Regarding adoption rates though, we do have metrics. PettingZoo is has been installed ~330k times on PyPI (e.g. through pip) in the year since it's public release and is the third most installed RL library through pip of all time, after Gym and Ray (which bundles RLlib): https://pepy.tech/project/gym, https://pepy.tech/project/rllib, https://pepy.tech/project/pettingzoo (note that these links are anonymized but link to sources which are not, please exercise caution).
>
> -Regarding biases in language against other models, that's a fair concern and we'll take a look at this when we address other comments on the paper.

---

### Decision · Program_Chairs · 2021-09-27

**Decision:**

Accept (Poster)

**Comment:**

I recommend this paper to be accepted. It provides a novel multi-agent reinforcement library with a corresponding games model. While the scientific novelty appears to be limited, the practical utility seems to be great for the community and such open-source libraries are explicitly encouraged in the Call for Papers.